# Strategic Approach to Massive Chylous Leakage after Neck Dissection

**DOI:** 10.3390/healthcare9040379

**Published:** 2021-03-31

**Authors:** Geng-He Chang, Chih-Yao Lee, Yao-Te Tsai, Chi-Cheng Fang, Ku-Hao Fang, Ming-Shao Tsai, Cheng-Ming Hsu, Chih-Wei Luan, Chang-Cheng Chang

**Affiliations:** 1Department of Otolaryngology-Head and Neck Surgery, Chang Gung Memorial Hospital, Puzi City 613, Taiwan; genghechang@gmail.com (G.-H.C.); yaote1215@gmail.com (Y.-T.T.); b87401061@cgmh.org.tw (M.-S.T.); scm00031@gmail.com (C.-M.H.); 2Graduate Institute of Clinical Medical Sciences, College of Medicine, Chang Gung University, Taoyuan City 333, Taiwan; jackluan2010@gmail.com; 3Health Information and Epidemiology Laboratory of Chang Gung Memorial Hospital, Puzi City 613, Taiwan; 4Department of Orthopaedics, Kaohsiung Medical University Hospital, Kaohsiung Medical University, Kaohsiung 807, Taiwan; yaoyao31430@gmail.com; 5School of Medicine, University of Queensland, Brisbane, QLD 4000, Australia; joe5836853@hotmail.com; 6Department of Otolaryngology-Head and Neck Surgery, Chang Gung Memorial Hospital, Taoyuan City 333, Taiwan; kuhaofang@gmail.com; 7Department of Otolaryngology-Head and Neck Surgery, Lo-Sheng Sanatorium and Hospital Ministry of Health and Welfare, New Taipei City 242, Taiwan; 8Department of Plastic and Reconstructive Surgery, China Medical University Hospital at Taichung, Taichung 404, Taiwan

**Keywords:** chyle, leak, fistula, cervical, pectoralis, cirrhosis

## Abstract

(1) Background: A high volume of chylous leakage (>1 L/day) is a potentially lethal complication after neck dissection. However, a strategic treatment for when the leakage progresses from high to massive (>4 L/day) is lacking. (2) Methods: The PubMed database was searched for articles on neck dissection–associated chylous leakage. Nine articles that included 14 cases with >1 L/day chylous leakage (CL) were analyzed. (3) Results: Of the nine patients with 1–4 L/day CL, three were successfully managed with conservative treatment, two with thoracic ductal ligation, three with ductal embolization, and one with local repair with a strap muscle flap. Of the remaining five cases with >4 L/day chylous leakage, three were successfully treated with the pectoralis major myocutaneous flap (PMMF) and one was successfully treated with thoracic ductal ligation and one case died. (4) Conclusions: In this review, when leakage was >4 L/day, the aforementioned interventions were ineffective, but applying the PMMF could rescue the intractable complication. We propose a strategic treatment for high (1–4 L/day) and massive (>4 L/day) chylous leakage.

## 1. Introduction

Patients with head and neck cancer often require neck dissection to reduce their risk of lymphatic metastasis. However, lymphatic dissection close to the lower neck may result in chylous leakage (CL), a complication that can cause nutrient loss, electrolyte imbalance, wound infection, and even death [1].

The management of CL is generally dependent on the daily amount of leakage [1]. Most scholars agree that for a low volume of CL, conservative management alone is effective, but for a high volume of CL, invasive intervention is required to control the leakage [2]. Most algorithms define the cutoff level between low and high volume as 1 L/day [2,3,4]. A study in 2000 by Nussenbaum et al. analyzed 635 neck dissections, of which 15 were complicated with CL (2.4%), and 4 of those CLs had peak daily amount >1 L/day (26.7%) [2].

Several invasive interventions are recommended for high CL volume (>1 L/day) or prolonged low CL volume (a duration >7 to 14 days) after conservative management has proven ineffective [2,4]. These interventions include lymphangiographic embolization of the thoracic duct [5,6,7], thoracoscopic thoracic duct ligation [3,4,8,9,10], exploratory surgical repair with fibrin glue [11], and the use of local rotational muscle flaps to obliterate the leakage sites [12].

In the literature, we found only studies that discussed treatment differences between low and high volumes of CL; however, cases with a massive CL volume, which poses considerable risk to the patient, are sometimes encountered in clinical settings. We speculated that this massive CL is an extreme condition and requires more aggressive treatment to reduce the risks of complication and mortality. To date, no research has investigated this extreme condition. Therefore, in this study, we reviewed the literature with a focus on massive CL. On the basis of the results of therapeutic interventions and outcomes of the collected cases, we attempted to define a cutoff value for “massive” leakage, and established a strategic management for this life-threatening complication.

## 2. Materials and Methods

A literature review was conducted by searching the PubMed database. The terms used for the search in subject headings and keywords were chyl*, lymphatic, thoracic duct, leak*, fistula*, cervical, neck, dissection, and surgery. The exact phrase in the search process was as follows: “(chyl* OR lymphatic OR thoracic duct) AND (leak* OR fistula*) AND (cervical OR neck) AND (dissection OR surgery).” The search was limited to publications from 1996 to 2018, studies enrolling subject headings of head and neck neoplasms. A total of 106 records were collected from the search process. We defined CL as a complication after cervical lymphatic dissection based on the subject and content of those articles.

Two reviewers (G.H.C. and J.C.F.) screened the 106 articles for eligibility. Initially, a general screening was performed that excluded articles on the basis of their titles and abstracts. Exclusion criteria were as follows: not within the topic of CL and neck dissection, fit the topic but did not focus on treatment, or not available in English. A total of 32 full-text articles were filtered for secondary screening. The main criterion for eligibility was inclusion of patients with CL of >1 L/day after a neck dissection and the CL should develop on the left side. The selection criteria did not discriminate in terms of articles from different countries, surgical interventions of any type, or treatment outcomes. In addition, other articles that were not discovered in the search but were found when reviewing the references of the search-identified articles and that met the criteria were included in the collection data. In total, nine studies were included in the final selection for analysis (Figure 1). Any disagreement during the screening was resolved through discussion or by consulting the third party (C.C.C.).

Data from each of the final nine studies comprised patient age, cancer type and stage, treatment (preoperative radiotherapy or other), presence of liver cirrhosis, surgery type, leakage site, maximum CL volume per day (defined as the highest reported CL volume in a single day), unsuccessful interventions, successful interventions, intervention times, resolution time, and final management results. We also recorded whether the patient survived after treatment, but for patients that died, we were unable to determine whether the cause of death was related to CL or other complications that occurred during CL management, such as stroke or myocardial infarction.

## 3. Results

In this review, nine articles containing 14 cases were included in the analysis (Table 1) [4,7,8,12,13,14,15,16,17]. The primary head and neck cancer of patients who underwent neck dissection included hypopharyngeal cancer (4/14, 29%), thyroid cancer (4/14, 29%), oral cancer (3/14, 21%), laryngeal cancer (1/14, 7%), melanoma (1/14, 7%), and unknown (1/14, 7%). The mean age of the patients was 60 (47–80) years. In patients with postoperative CL, the main types of neck dissection were modified radical neck dissection (the spinal accessory nerve (SAN), internal jugular vein (IJV), or sternocleidomastoid muscle (SCM) may be retained) (5/14, 35.7%) and radical neck dissection (the aforementioned three are not retained) (5/14, 35.7%). The rest included three selective neck dissections (3/14, 21.4%) (to remove lymph nodes in selective regions) and one functional neck dissection (1/14, 7.1%) (to preserve SAN, IJV, and SCM). Among the 14 cases, seven nodal stages were unknown, and of the other seven known nodal stages, four were advanced (two N2 and two N3) and three were early (N1). The advanced nodal stages (N2 or N3) all occurred in cases with >4 L/day CL, and the nodal stages of cases with <4 L/day CL were all early (N1). Of the 14 cases, nine had a CL volume of 1–4 L/day and the other five had a CL volume of >4 L/day. 

Of the nine patients with a CL volume of 1–4 L/day, seven were initially treated with conservative management, including bed rest, pressure dressings on neck, lower pressure suction drainage, enteral feeding with medium-chain triglycerides (MCTs), and fasting with total parenteral nutrition (TPN) replacement, which was successful in three patients but not in four patients. In other words, conservative treatment for 1–4 L/day CL had a failure rate of 57%. The remaining four patients, who were refractory to conservative treatment, and another patient who was initially unsuccessfully treated with local repair, were successfully treated with thoracic ductal ligation (2/5), ductal embolization (2/5), and surgical repair (1/5).

In the five patients with >4 L/day CL, three were successfully treated using the pectoralis major myocutaneous flap (PMMF). The CL volumes of the three patients were 5.93, 4.30, and 4.03 L/day, and the success rate of PMMF in these patients was 60%. The interval between CL initiation and PMMF surgery was 21, 28, and 18 days, respectively. The time for complete remission of CL after surgery was 8, 1, and 18 days, respectively. 

A patient with a CL volume of >7 L/day died after surgical repair and thoracic ductal ligation. Liver cirrhosis was identified in this patient and in another patient with a CL volume of 5.93 L/day who was treated using PMMF after failure of surgical repair (5.93 L/day). In addition, three of the patients with a CL volume of >4 L/day did not undergo preoperative radiotherapy (3/5, 60%).

Of the five cases with CL > 4 L/day, two cases were directly and successfully treated using the pectoralis major myocutaneous flap (PMMF) and fibrin glue, and one case failed with an SCM flap after which they were rescued using the PMMF and fibrin glue. The time of the PMMF procedure was the 21st, 28th, and 18th day after neck dissection, respectively, and the CL was resolved at 8, 1, and 18 days postoperation. Thoracoscopic thoracic ductal ligation was employed to directly terminate another massive leakage; however, local repair and thoracic ductal ligation failed to control a > 7L/day CL and resulted in death. In summary, massive CL led to high mortality (1/5, 20%), and the survivors were mostly rescued using the PMMF (3/4, 75%).

## 4. Discussion

Neck-dissection-related CL is uncommon, but CL of more than 1 L/day can result in a life-threatening complication (mortality rate for >1 L/day CL group: 1/14, 7.14%; >4 L/day CL group: 1/5, 20%). The terminus of the thoracic duct is usually located on the left lower neck and has multiple fragile branches that drain 3 to 5 L of lymph per day [6]. Anatomic studies have discovered considerable variation in the termination [18,19]. Kinnaert et al. reported that only 13% of ducts terminate into the venous system through a single communication, and the others had multiple endings [20]. Therefore, CL is difficult to prevent completely and has 1% to 2.5% incidence, even when the operation is performed by an experienced surgeon [2]. 

Head and neck cancers often require neck dissection to decrease the risk of metastasis and local recurrence of cervical lymph nodes [1,2]. In general, preventive neck dissection involves levels I–III of the neck, which is the supraomohyoid region and includes the omohyoid muscle at the bottom up to the mandible [6]. Occasionally, for positive lymph nodes and advanced laryngeal or hypopharyngeal cancer, the areas of neck dissection may include level IV, which is located between the omohyoid muscle and the clavicle and is the site where the thoracic duct enters the subclavian vein [6]. Herein, the inclusion of level IV in neck dissection is more likely to be associated with CL.

According to the results of previous studies, a CL volume of 1 L/day is often used as a cutoff value to classify CL volumes as low and high [2,3,4]. We defined massive CL as that with a volume of >4 L/day based on our review results. We observed that the effective treatments for volumes below and above 4 L/day of CL were different. PMMF is a good therapeutic method when the leakage exceeded 4 L/day, and three of the five patients with >4 L/day of CL were successfully treated with PMMF. In addition, one of the three patients was successfully treated with PMMF after failure of traditional surgical repair with sternocleidomastoid muscle. Therefore, we believe that the most effective management strategy varies for CL above and below 4 L/day and we selected 4 L/day as the cutoff for massive CL and divided CL into high and massive groups to devise a therapeutic strategy for CL management.

### 4.1. High Volume of CL: 1–4 L/Day

Exploratory surgical repair is generally accepted and remains the mainstay treatment for high CL volume (>1 L/day). De Gier et al. [12], one of our included studies, reported three cases with CL of 2.1, 2.3, and 2.5 L/day, respectively, that were eventually terminated through conservative management alone; however, the duration of resolution was 3 to 4 weeks. Although 2 to 3 L/day CL was successfully treated using conservative management in de Gier et al., the prolonged period of resolution may have increased the risk of developing formidable complications. Conversely, conservative therapy failed to treat four cases of 1–4 L/day CL included in the review (3.4, 3.3, 2.1, and 2.5 L/day, respectively, in Gunnlaugsson et al. [4], Van Goor et al. [7], Chen et al. [14], and Casler et al. [13]). Consequently, we conclude that conservative management should remain the mainstay choice for a low volume of CL (<1 L/day) but not >1 L/day leakage.

Alejandre-Lafont et al. [5] introduced therapeutic lymphography for thoracic ductal embolization and discovered it could detect 80% of leakage sites. However, as Alejandre-Lafont et al. reported, the success rate was 70% for <500 mL/day CL but only 35% for >500 mL/day CL. In fact, of the nine cases with 1–4 L/day CL in our review, thoracic ductal embolization successfully terminated CL in three cases, including two for which conservative therapy had failed. Therefore, we propose that thoracic ductal embolization, the minimally invasive procedure, be considered for <1 L and 1–4 L CL as an alternative, if available.

Thoracoscopic thoracic ductal ligation has also been proven to be an efficient and minimally invasive surgical treatment for >1 L/day CL [3,4,8,9,10]. Gunnlaugsson et al. [4] and Ilczyszyn et al. [8] reported two cases with CL of 2.8 and 3–4 L/day. The two cases were refractory to both conservative management and exploratory surgical repair with a local SCM flap but use of thoracic ductal ligation eventually terminated the leakage with rapid resolution of 2 and 5 days, respectively. In addition, Wilkerson et al. [16] reported that the procedure successfully rescued 5 L/day CL with rapid resolution in 3 days. Thoracoscopic surgery appears to be effective, even when CL volume is 4 L/day. Therefore, besides exploratory surgical repair, thoracoscopic thoracic ductal ligation can be considered an alternative or prior choice for the management of 1–4 L/day leakage.

The analysis of CL volumes of 1–4 L/day indicates that opening the wound to repair the leakage site is the mainstay of treatment. However, in some cases, such as when the muscle around the wound is excised during neck dissection and no sufficient material is available for repair, the wound is unsuitable for reopening and physicians can consider ductal ligation; alternatively, when leakage persists but does not exceed 4 L/day after surgical repair, then physicians can consider thoracic ductal ligation. Ductal embolization can also be applied in some cases, such as when the patient’s condition does not allow for reoperation, when surgical repair fails but the leakage volume does not exceed 4 L/day, or when the leakage volume is initially low and physicians aim to manage CL without surgical treatment. In summary, surgical repair can be the primary treatment choice, but ductal ligation or embolization can also be regarded as alternatives under some circumstances.

### 4.2. Massive Volume of CL: >4 L/Day

In our review, five patients had CL volumes of >4 (5–7) L/day. Surgical repair with a local flap failed to control the massive CL in two patients. A patient with 5 L/day CL was successfully treated with thoracic ductal ligation, but the same procedure failed in another patient with a CL volume of >7 L/day. By contrast, PMMF successfully controlled three massive CLs (5.93, 4.30, and 4.03 L/day). However, PMMF use was delayed in these three patients at 3–4 weeks into the intervention (18th, 21st, and 28th days). Consequently, PMMF can be thought of as a salvage procedure and an effective treatment for intractable CL (>4 L/day) and potentially life-threatening conditions. When the CL volume exceeds 4 L/day, surgery using PMMF should be immediately considered to effectively resolve the leakage as soon as possible and reduce the risk of complications and mortality.

### 4.3. Pectoralis Major Myocutaneous Flap

Exploratory surgical repair by ligation of the damaged duct can be difficult because of the multiple branches, structural variation, and fragile nature of the thoracic duct. The surrounding local fascia or muscle can be rotated to cover the defect, but this is sometimes impossible because the muscle is insufficient or was sacrificed in a radical neck dissection. The bulky PMMF is proximal to the lower neck and can be easily accessed to obliterate the space, forming a strong and efficient tamponade barrier that seals off all leaking branches of the thoracic duct and terminates intractable CL. In addition, it can be harvested using a skin paddle to resurface the skin defect if necrotic skin is sacrificed during the procedure. Among the four survivors with >4 L/day CL in our review, the PMMF rescued three cases (3/4, 75%) without failure. Even though microsurgical free tissue transfer is frequently used in reconstruction during head and neck surgery, the PMMF is a useful alternative when free flap transfer is not suitable for the primary or salvaging surgery.

### 4.4. Association with Liver Cirrhosis

Parasher et al. [21] employed endoscopic ultrasound to discover considerable thoracic duct dilatation in patients with liver cirrhosis or portal hypertension. Dumont and Mulholland [22,23] demonstrated a 3- to 6-fold increased flow rate and significantly raised pressure of the thoracic duct in patients with liver cirrhosis. High ductal flow and pressure-deteriorating chylothorax have been reported to become a life-threatening condition in patients with liver cirrhosis, and treatment of portal hypertension using a transjugular intrahepatic portosystemic shunt has been investigated and determined to be useful in the refractory chylothorax and ascites [24,25,26]. In addition, according to Eufinger and Lehmbrock [15], comorbid liver cirrhosis could be a reason for extreme CL (>7 L/day) in one patient who died. In our review, among the three cases with >5 L/day CL [15,16,17], two cases (2/3, 67%) had liver cirrhosis. In conclusion, liver cirrhosis may be a factor that contributes to the development of a massive and life-threatening CL. Therefore, screening the liver condition is warranted when leakage is massive or intractable.

### 4.5. The Strategic Approach

Previous studies suggest conservative management to be the first-line approach when CL volume is low (<1 L/day). Some active interventions are recommended when the conservative treatment fails, or the leaking amount arrives at high level (>1 L/day). However, a consensus is lacking about the timing and interventive steps for systemically managing the leakage, especially when it arrives at massive amount (>4 L/day). Therefore, based on the study, we propose an algorithm for CL management that highlights the treatment of massive CL and extends the indication for the PMMF as a salvaging intervention (Figure 2).

CL should be carefully identified and managed through direct repair during operations. If the leak develops postoperatively, the management should be based on the daily amount of leakage. Based on our global review, for high CL (1–4 L/day) or prolonged low CL after conservative management for 7–14 days, exploratory surgery is the mainstay treatment for stopping the leakage. Alternatively, if available, thoracic ductal embolization or thoracoscopic thoracic ductal ligation can be considered as a minimally invasive choice. When the CL is >4 L/day or the aforementioned interventions fail to treat a leakage, application of the PMMF is believed to be an efficient therapy for rescuing patients from life-threatening complications.

Our studies reviewed the literature to systematically analyze the management of chylous leak after neck dissection and propose a therapeutic algorithm. There might be some potential biases and limitations. Mainly, high and massive chylous leak is a rare complication for reports and thus, only a small number of cases could support the proposed algorithm. However, our study offers potentially useful guidance when physicians encounter life-threatening CL. We applied the Newcastle–Ottawa scale (NOS) to evaluate the quality of these studies [27]. NOS results of 6 points or more belong to high-quality studies. The results showed that the cases that we included were all high quality with 6 points of NOS (Appendix A).

## 5. Conclusions

Massive CL is a clinically inevitable and severe problem, but research on its definition and relevant management is lacking. On the basis of our review, we defined cutoff values and proposed a strategic algorithm for managing massive (>4 L/day) amount of CL. When the CL is >4 L/day, the use of the PMMF should be considered in a timely manner as a rescuing procedure to control and terminate the intractable and life-threatening complication.

## Figures and Tables

**Figure 1 healthcare-09-00379-f001:**
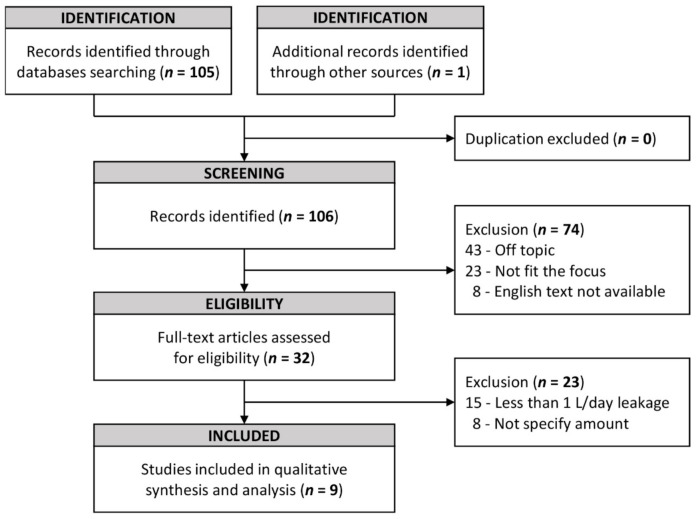
PRISMA (Preferred Reporting Items for Systematic Reviews and Meta-Analyses) flow diagram of article enrollment.

**Figure 2 healthcare-09-00379-f002:**
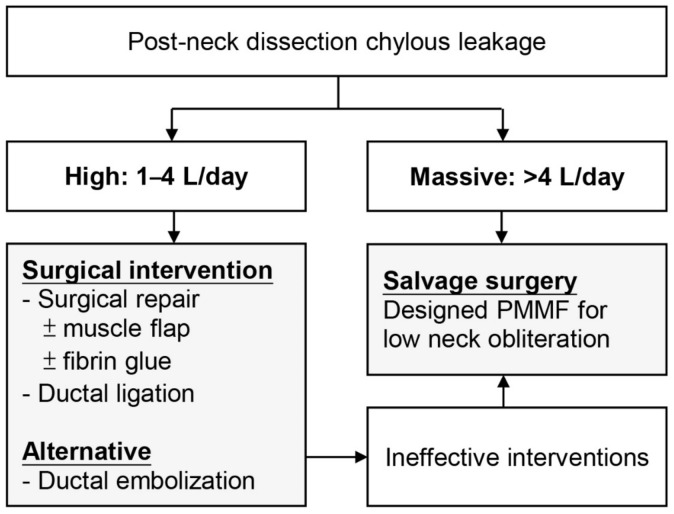
Strategic approach for neck-surgery-related chylous leakage. CL: chylous leakage; SCM: sternocleidomastoid muscle flap; strap: strap muscle flap; MCT: medium-chain triglycerides diet; TPN: total parenteral nutrition; PMMF: pectoralis major myocutaneous flap; Ductal embolization: lymphangiography for thoracic ductal embolization; Ductal ligation: thoracoscopy for thoracic ductal ligation.

**Table 1 healthcare-09-00379-t001:** Studies including cases with high and massive volume of chylous leakage in a global review.

Author	Year	Age	Cancer	Stage	Pre-OP RT	Liver Cirrhosis	OP	Side	Maximal Amount	Failed Intervention	Successful Intervention	Time of OP	Time to Resolve	Outcome
Eufinger	2001	66	Unknown	Unknown	Unknown	Yes	FND	L	7 L/day	Repair/Ligation	Nil	Nil	Nil	Death
Su	2017	54	HPX	T4N2	No	Yes	MRND	L	5.93 L/day	Repair (SCM)	PMMF	21st day	8 days	Survivor
de Gier	1996	51	Larynx	T3N3	No	Unknown	RND	L	4.30 L/day		PMMF	28th day	1 day	Survivor
de Gier	1996	76	HPX	T4N3	No	Unknown	RND	L	4.03 L/day		PMMF	18th day	18 days	Survivor
Wilkerson	2013	60	HPX	T2N2b	Yes	Unknown	MRND	L	5.00 L/day		Ligation *	6th day	3 days	Survivor
Gunnlaugsson	2004	49	TON	T2N1	Unknown	Unknown	SND	L	3–4 L/day	Conservation	Ligation	4th day	5 days	Survivor
Ilczyszyn	2011	80	Gum	T4N1	Unknown	Unknown	MRND	L	2.89 L/day	Repair (SCM)	Ligation	11th day	2 days	Survivor
de Gier	1996	49	MTC	Unknown	No	Unknown	RND	L	2.53 L/day		Conservation *		21 days	Survivor
de Gier	1996	73	Melanoma	Unknown	No	Unknown	MRND	L	2.36 L/day		Conservation		30 days	Survivor
de Gier	1996	63	MTC	Unknown	No	Unknown	MRND	L	2.17 L/day		Conservation		30 days	Survivor
Van Goor	2007	55	PTC	Unknown	Unknown	Unknown	SND	L	3.30 L/day	Conservation	Embolization *	20th day	5 days	Survivor
Van Goor	2007	63	MTC	Unknown	Unknown	Unknown	SND	L	2.40 L/day		Embolization	13th day	1 day	Survivor
Chen	2016	51	Tongue	T1N1	Yes	Unknown	RND	L	2.10 L/day	Conservation	Embolization	23rd day	3 days	Survivor
Casler	1998	47	HPX	Unknown	Unknown	Unknown	RND	L	2.50 L/day	Conservation	Repair *(strap)	28th day	Unknown	Survivor

RT: radiotherapy; OP: operation; CL: chylous leakage; FND: functional neck dissection; HPX: hypopharynx; MRND: modified radical neck dissection; SCM: sternocleidomastoid muscle flap; PMMF: pectoralis major myocutaneous flap; RND: radical neck dissection; TON: tonsil; SND: selective neck dissection; PTC: papillary thyroid cancer; MTC: medullary thyroid carcinoma; Ligation *: thoracoscopy for thoracic ductal ligation; Embolization*: lymphangiography for thoracic ductal embolization; Repair *: reopen the wound and repair the leaking site; strap muscle flap; Conservation *: conservative therapies.

## Data Availability

The data and statistical analysis information used in this research have been fully presented in the article.

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
