# Peer review of "Strategic Approach to Massive Chylous Leakage after Neck Dissection"

_healthcare, 2021, doi:10.3390/healthcare9040379_

Round 1
Reviewer 1 Report
This paper is the study about “Strategic approach to low (<1 L/day), high (1–4 L/day), and massive (>4 L/day) chylous leakage after neck dissection”. We think that this study is interesting.
However, I made the same point last time, the study examined was only ‘Massive’, with no validation for other levels (low and high). And the title is questionable. Please make the purpose of this study clear and consider reconsideration.
Some comments are as below:
- I have repeatedly pointed out the title and content, but it has not been fixed.
Your systematic review is only "Massive ".
It is strange to treat “Low” and “High” equally.
- Describe the reason for the CL threshold.
What is the definition of Massive? 
It is said that it is based on the results of your review, but the content is not written on Introduction and discussion.
I want a logical development again.
Reviewer 2 Report
Now the authors have definitely improved their manuscript, congratulations!
Author Response
Thank you much for your advice given at last time and affirmation, and thank you again for your time and consideration in reviewing this manuscript.
Reviewer 3 Report
Dear Editor and Authors, thanks for inviting me to review this well-written manuscript.
I want to send my congratulations to the authors. Because despite the low number of cases, they perform a really nice review and propose a treatment algorithm really useful.
Author Response
Thank you much for your valuable suggestions last time so that we can improve our article, and thank you again for your time and consideration in reviewing this manuscript.
Reviewer 4 Report
This is an interesting review about approaches to low, high, and massive chylous leakage after neck dissection. The authors found Nine articles that included 14 cases with >1 L/day chylous leakage. They analyzed the results from literature and proposed an algorithm for treatment. PRISMA guidelines were correctly used for the systematic review.
The paper is well written. However, some issues remain.
In the Results, the type of neck dissection was reported only for 10/14 patients. What about the other 4 patients?
Please summarize the node stage results in the text.
The authors should better specify what “conservative treatment” means not only in the Discussion, but also in the Results.
There was a correlation between chylous leakage volume and type of neck dissection? Moreover, it may be interesting to report is there was a correlation between chylous leakage volume and conservative and/or non-conservative treatment success.
In the Discussion, please specify that thoracic duct is usually on the left side of the neck.
Reviewer 5 Report
The Authors present an interesting paper with an actual topic regarding chylous leakage afte neck dissection. According to the literature, the Authors don't add any new concept.
Author Response
Thank you much for your valuable suggestions last time, which gave us the opportunity to improve our article, and thank you again for your time and consideration in reviewing this manuscript.
Round 2
Reviewer 1 Report
This paper is the study about “Strategic approach to massive chylous leakage after neck dissection”. We think that this study is interesting.
I think that the pointed out matter has not been corrected, and cannot be accepted. The content needs to be revised significantly. Please make the purpose of this study clear and consider reconsideration.
Some comments are as below:
- I have repeatedly pointed out the title and content, but it has not been fixed.
Your systematic review is only "Massive ".
It is strange to treat “Low” and “High” equally.
- Describe the reason for the CL threshold.
What is the definition of Massive? 
It is said that it is based on the results of your review, but the content is not written on Introduction and discussion.
I want a logical development again.
Author Response
Thank you much for your comment and give us another precious opportunity to reply to your question.
#1. I have repeatedly pointed out the title and content, but it has not been fixed.Your systematic review is only "Massive ". It is strange to treat “Low” and “High” equally.
-- Response --
Thank you much for raising this important question again. Because our systemic review mainly discusses the definition and disposal of massive chylous leakage, we have modified the title and content in our last reply, emphasizing that this article is specifically for massive chylous leakage.
We included low and high treatment recommendations in the discussion, mainly because the treatment of chylous leakage varies according to the increase in volume, we need to describe a staged treatment, and then we could emphasize the difference and importance of the treatment for massive chylous leakage from that of low and high chylous leakages.
In the part, conservative treatment has been determined to be the first choice for low chylous leakage (<1L/day). We just made descriptive discussions based on past research findings and clinical experience for the low chylous leakage.
In addition, regarding the treatment of high chylous leakage (>1L/day), past research and clinical experience recommend intensive treatment. Therefore, we coordinated with the cases in the article to organize and define it as 1-4L/day and gave advice on disposal.
With the definition and treatment of the first two levels, we could clearly describe the volume of massive chylous leakage, and how the treatment is different from the two. We need such a staged description to make the article clearly express the definition of massive chylous leakage and the recommended treatment.
#2. Describe the reason for the CL threshold. What is the definition of Massive? It is said that it is based on the results of your review, but the content is not written on Introduction and discussion. I want a logical development again.
-- Response --
Thank you much for pointing out this very important issue again. We defined 4L/day as the cuff-off value between massive and high chylous leakage, and believe that leakage of more than 4L/day requires more aggressive action to use PMMF for treatment.
We defined this way mainly based on the results of our review, which was also the main purpose of our research. Among the cases we collected, we found that there was an obvious difference between the effective therapeutic methods of leakage above and below 4L/day. Leakage above 4L/day was mainly effectively treated with PMMF, and the success rate was high. Therefore, based on this result, we used 4L/day as a cut-off value to define the massive chylous leakage and established a strategic approach for managing chylous leakage.
How we chose this cuff-off value and reasons were explained in the introduction and discussion. [line 58-63 for introduction; line 156-164]
We hope that our reply will satisfy you, and thank you again for taking the time to review.
Reviewer 5 Report
The paper is not so impressive to be published
Author Response
Thank you much for your comment. Although the number of articles and cases collected by our review is small, we believe that the findings of our article are valuable for clinical application.
In the past, for the treatment of chylous leakage, it is clear that 1L/day is used to define the difference between low and high leakage, and it is recommended that conservative management is for low leakage and invasive treatment is for high volume of leakage, mainly because many studies have found that such management is reasonable and can successfully control and treat most chylous leakage.
However, there was no research to explore the state of "massive" volume in the past. We have encountered massive chylous leakage clinically and believe that there is an extreme state that requires more aggressive treatment to effectively control the life-threatening complication. Therefore, we reviewed past articles including cases with high chylous leakage and tried to explore whether the massive leakage exist.
The results of the analysis allowed us to find that when the leakage exceeded 4L/day, the effective treatment was almost PMMF, which was very different from when the leakage was less than 4L/day. Therefore, we defined the leakage exceeding 4L/day as a massive chylous leakage and recommend that the effective therapeutic modality is PMMF. We are the first to establish the definition for “massive” chylous leakage and also the first to recommend that PMMF is the choice for managing the extremely high volume of leakage.
Perhaps in the past, PMMF was intuitively used to treat an extreme leakage, but there was no academic evidence to prove the treatment is effective and should be considered timely. Therefore, we believe that our research is of clinical value. First, we conducted the review to logically find a cut-off value to define the massive chylous leakage. Next, we have verified that PMMF is indeed an effective treatment for this extremely dangerous situation. Finally, the methods for treating low and high chylous leakage, which have been widely used clinically were integrated with our findings to establish a strategic approach, which could help clinicians have a more comprehensive guidelines when treating the neck dissection-associated complication.
We hope that the above responses to elaborate the value of our article would satisfy you, and thank you again for taking the time to review our article and giving us the opportunity to expound it again.
This manuscript is a resubmission of an earlier submission. The following is a list of the peer review reports and author responses from that submission.
Round 1
Reviewer 1 Report
This paper is the study about “Strategic approach to low (<1 L/day), high (1–4 L/day), and massive (>4 L/day) chylous leakage after neck dissection”. We think that this study is interesting.
However, the only study examined was ‘Massive’, with no validation for other levels (low and high). And the classification of the data is questionable. There is little discussion for Massive, and the format is questionable. Please make the purpose of this study clear and consider reconsideration.
Some comments are as below:
- Describe the background to CL
- Describe the reason for the CL threshold
- Description of frequency for the amount of CL
- Describe the Level classification for the extent of cervical dissection.
Author Response
Dear Reviewer:
We are happy to learn that our manuscript “Strategic approach to low (<1 L/day), high (1–4 L/day), and massive (>4 L/day) chylous leakage after neck dissection” had been reviewed by Healthcare. We greatly appreciate your consideration and have amended the manuscript according to your suggestions. Our full responses to the comments are attached below, and the corresponding changes are shown in red in the revised manuscript.
Chang-Cheng Chang, MD, PhD (Corresponding author)
Department of Plastic and Reconstructive Surgery, China Medical University Hospital at Taichung, Taiwan
Phone: 886-4-2205-2121 ext. 2020
E-mail: a9244@adm.cgmh.org.tw

Reviewer 2 Report
The paper entitiled "Strategic approach to low (<1 L/day), high (1–4 L/day), and massive (>4 L/day) chylous leakage after neck dissection" offers useful information for readers who are intersted in this field.
I think this paper is well documented about treatment for massive chylous leakage. However, let me say something. The authors mention thoracic ductal embolization or thoracoscopic ductal ligation can be considered for high CL(1-4L/day) as a minimally invasive choice. Please clarify the indications for these treatment instead of conventional surgical repair. Also, on Figure2, ductal ligation is classified as alternative treatment. Should it be classified as surgical repair?Author Response
Dear Reviewer:
We are happy to learn that our manuscript “Strategic approach to low (<1 L/day), high (1–4 L/day), and massive (>4 L/day) chylous leakage after neck dissection” had been reviewed by Healthcare. We greatly appreciate your consideration and have amended the manuscript according to your suggestions. Our full responses to the comments are attached below, and the corresponding changes are shown in red in the revised manuscript.
Chang-Cheng Chang, MD, PhD (Corresponding author)
Department of Plastic and Reconstructive Surgery, China Medical University Hospital at Taichung, Taiwan
Phone: 886-4-2205-2121 ext. 2020
E-mail: a9244@adm.cgmh.org.tw

Reviewer 3 Report
Overall a quick review dealing with a rare complication of HN surgery.
Abstract: pubmed= medline, you don't "enroll" an article (include) >>> have the whole text revised by a native English speaker!
Introduction: needs to be reformatted, please clearly identify the key premises of your work, line 47 advanced >> invasive?
MM: why older than 45? such an exclusion criterion does not make any sense to me, PICCO criteria are unclear: how was CL defined/measured? is death clearly related to CL? what do you mean by conservative? (total parenteral nutrition, dressings, somatostatin?) how many days you have to wait before measuring CL volume? Bias and quality of studies was not assessed: why? sometimes a narrative review is a better format...
Results: presentation in confusing, especially in the first paragraphs. Mean 60 years: did you mean median? 100% left side: this should be an inclusion criteria IMO.
Discussion: the flow chart and discussion are fine besides some small errors. However, this "systematic review" is actually a case-series of only 14 patients: a critical discussion of the management of high volume CL would have served better to the readers.
Author Response

(The authors gave the same response as above.)

Round 2
Reviewer 1 Report
This paper is the study about “Strategic approach to low (<1 L/day), high (1–4 L/day), and massive (>4 L/day) chylous leakage after neck dissection”. We think that this study is interesting.
However, I made the same point last time, the study examined was only ‘Massive’, with no validation for other levels (low and high). And the title is questionable. Please make the purpose of this study clear and consider reconsideration.
Some comments are as below:
- Abbreviations are not written in the text.
- Describe the reason for the CL threshold. What is the definition of Massive? I want a logical development.
Reviewer 3 Report
Fine for your comments and amendments but still I keep my doubts on the format: a critical/narrative review would probably have served better than a meta-analysis. I still recommend a check with a native English speaker: for instance, line 30 we search PubMed, line 33 cases is repeated twice, 44-45 lymphatic > there are synonyms, line 46 would > may in the figure, salvaging surgery> salvage surgery.... etc.